# InfoRank: Unbiased Learning-to-Rank via Conditional Mutual Information Minimization

## ABSTRACT

Ranking items regarding individual user interests is a core technique of multiple downstream tasks such as recommender systems. Learning such a personalized ranker typically relies on the implicit feedback from users' past click-through behaviors. However, collected feedback is biased toward previously highly-ranked items and directly learning from it would result in "rich-get-richer" phenomena. In this paper, we propose a simple yet sufficient unbiased learning-to-rank paradigm named InfoRank that aims to simultaneously address both position and popularity biases. We begin by consolidating the impacts of those biases into a single *observation* factor, thereby providing a unified approach to addressing bias-related issues. Subsequently, we minimize the mutual information between the *observation* estimation and the *relevance* estimation conditioned on the input features. By doing so, our relevance estimation can be proved to be free of bias. To implement InfoRank, we first incorporate an attention mechanism to capture latent correlations within user-item features, thereby generating estimations of observation and relevance. We then introduce a regularization term, grounded in conditional mutual information, to promote conditional independence between relevance estimation and observation estimation. Experimental evaluations conducted across three extensive recommendation and search datasets reveal that InfoRank learns more precise and unbiased ranking strategies.

## 1 INTRODUCTION

Implicit feedback garnered from user interactions (such as clicks) serves as a prevalent data source in situations where "ground truth" about explicit relevance is difficult to obtain on a Web platform. Although implicit feedback offers the advantage of mitigating data labeling expenses, it introduces a spectrum of bias concerns [4, 25]. As summarized in [10], two typical forms of bias emerge during the data collection and serving phases: position bias and popularity bias. These biases would be exacerbated by the feedback loops inherent in ranking systems, as depicted in Figure 1(a). Concretely, position bias manifests during the collection of user feedback and arises due to user browsing patterns. As Figure 1(b) shows, users typically peruse presented item lists from top to bottom, with their attention diminishing rapidly along the way [24, 25]. Consequently, higher-ranked items receive more exposure and greater opportunities for *observation* and subsequent clicks [14, 15], leading to

Permission to make digital or hard copies of all or part of this work for personal or classroom use is granted without fee provided that copies are not made or distributed for profit or commercial advantage and that copies bear this notice and the full citation on the first page. Copyrights for components of this work owned by others than ACM must be honored. Abstracting with credit is permitted. To copy otherwise, or republish, to post on servers or to redistribute to lists, requires prior specific permission and/or a fee. Request permissions from permissions@acm.org.

*WWW'24, May 13 - 17, 2024, Singapore*

© 2023 Association for Computing Machinery.

ACM ISBN 978-x-xxxx-xxxx-x/YY/MM...$15.00

https://doi.org/10.1145/nnnnnnn.nnnnnnn

an increased possibility of being collected as implicit feedback. In contrast, popularity bias occurs when the system returns ranked lists for user service, as illustrated in Figure 1(b). This bias prompts ranking systems to recommend popular items more frequently than their popularity would warrant. In both cases, blindly optimizing ranking performance based on implicit feedback data may inadvertently reinforce the existing presentation or popularity order rather than learning personalized relevance, as higher-ranked or popular items naturally garner more user observations and increased opportunities to elicit positive feedback such as clicks.

Recently, numerous studies have raised awareness of such bias issues and have strived to uncover the underlying relevance from such biased feedback. To address position bias, recent research has harnessed counterfactual learning technique [31], wherein the position is treated as a counterfactual factor [37], and employing inverse propensity weighting (IPW) [2, 23, 49] to rectify user feedback. In the context of popularity bias, recent endeavors have explored diverse strategies, including diversity-based regularization [1, 28] to balance rating and viewpoint values, and adversarial learning approaches [29] designed to uncover implicit associations between popular and less popular items. However, it is noteworthy that these methods often focus on either position or popularity factors in isolation, and lack a unified debiasing framework.

In this paper, we reveal that these prevalent biases, which manifest during either the data collection or serving phases, exert their influence on ranking performance through what we term the "observation" factor. For example, position bias originates from the user perspective, as the item's position correlates with whether it has been observed, thereby influencing user feedback; and popularity bias is induced by the ranking system, as the system-generated list is influenced by the frequency of user observations and subsequent clicks on a given item. This insight motivates us to estimate user-item relevance free from the effect of the observation factor, since our premise is that user-item features should be the sole determinants of relevance. In other words, irrespective of an item's position or the number of times it has been observed previously, its relevance to the user should remain consistent.

To this end, we propose InfoRank, a novel unbiased learning-to-rank paradigm that allows us to model the relevance from rich user-item profiles while mitigating the biases raised by the observation factor. Our approach begins with the incorporation of an attention mechanism to capture concealed correlations embedded within the rich user-item features. To illustrate, consider the scenario depicted in Figure 1(c): Imagine a teenager searching for shoes, and the ranking system subsequently presents several shoe options based on her profile and historical data. If, for instance, she has a limited budget (a user feature), she is more inclined to favor shoes with lower prices (an item feature). These correlations hold significant importance in uncovering users' preferences concerning the presented items. To ensure that our estimated relevance is solely

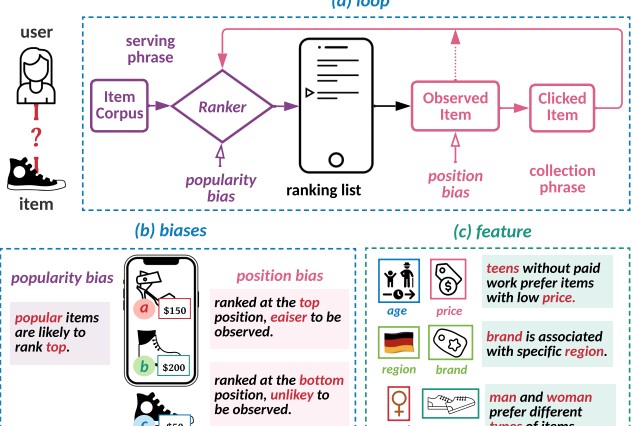

**Figure 1: An illustrated example of the feedback loop, position bias, and popularity bias in learning-to-rank. Within this process, the ranking system blends user and item features (c) with implicit feedback to generate the final ranking list. However, this system is susceptible to both position bias and popularity bias (b). Furthermore, these biases tend to be amplified within the feedback loop (a), potentially resulting in a "rich-get-richer" dilemma.**

influenced by user-item features, we introduce the *observation* factor as a latent variable. Then, we encourage the unbiased estimation (i.e., relevance) to be independent of the biased factor (i.e., observation) conditioned on the input features by minimizing their mutual information. We further derive a regularization formulation of the conditional mutual information minimization. Correspondingly, we devise a novel end-to-end framework that concurrently optimizes ranking performance while liberating the ranker from position and popularity biases stemming from implicit feedback.

Experiments conducted on three diverse datasets, demonstrate the superiority of InfoRank when compared to the state-of-the-art baselines across a variety of browsing patterns. Moreover, our ablation studies illustrate that our conditional mutual information minimization has the potential to enhance the performance of other ranking models as well.

## 2 BRIDGING BIASES IN RANKING TO DEPENDENCE IN CAUSALITY

### 2.1 Biased and Unbiased Learning-to-Rank

The objective of a point-wise learning-to-rank algorithm is to train a ranker $f$ that assigns a relevance score to each query-item pair (or referred to as a user-item pair). However, as explicit relevance signals are usually too expensive in practice, biased ranking algorithms directly replace them with implicit click signals. Let $u$ denote the user and $d$ denote the item. Let $\boldsymbol{x}$ denote the feature vector of $u$ and $d$. Let $\mathcal{D}$ denote the whole set of items and $\mathcal{D}_u$ denote the set of items associated with $u$. $c$ represents whether $d$ is clicked or not. The risk function in learning is defined as

$$\mathcal{F}(f) = \sum_u \sum_{d \in \mathcal{D}_u} \Delta(f(\boldsymbol{x}), c), \qquad (1)$$

where $\Delta(f(\boldsymbol{x}), c)$ denotes a point-wise loss function.

**Table 1: A summary of notations.**

| Notations | Explanations |
|---|---|
| $O, C$ | Observation, click attributes (implicit feedback) |
| $X = (U, I, P)$ | Features (user and item side including position) |
| $R$ | Relevance attribute (unbiased estimation) |
| $\mathcal{I}$ | Conditional mutual information, see Eq. (13) |
| $\mathcal{D}, \mathcal{D}_u$ | Dataset, dataset associated with user $u$ |

Traditionally, ranker models are trained using user browsing logs including labeled click data to find the optimal ranker that minimizes the risk function. However, this conventional approach is vulnerable to both position bias [4, 27] and popularity bias [1, 28]. We refer to this as biased learning-to-rank. Recent endeavors have introduced unbiased learning-to-rank algorithms aiming to eliminate such biases in click data and train an unbiased ranker.

**Position Bias.** Many prior studies [4, 22, 24, 49, 50] have underscored the observation that items occupying higher positions are more prone to being both observed and subsequently clicked. Consequently, training a ranker directly on click data may lead to it primarily estimating the position order rather than the personalized relevance of items. To rectify this inherent bias, conventional debiasing methods typically introduce an additional relevance factor, denoted as $r$, to indicate the relevance of item $d$.[*] Then, these methods estimate $r$ instead of $c$ in ranking. To this end, they leverage the insight that a user clicks on an item only when it has been both observed and perceived as relevant. This relationship can be formulated as:

$$P(C = 1|X = \boldsymbol{x}) = P(R = 1|X = \boldsymbol{x}) \cdot P(O = 1|X = \boldsymbol{x}). \qquad (2)$$

where $C, R, O, X$ denote the random variable.[†] Then, the objective of unbiased learning-to-rank is to infer relevance from click data and generate a ranked list based on $P(R = 1|X = \boldsymbol{x})$ different from biased learning-to-ranking using $P(C = 1|X = \boldsymbol{x})$.

**Popularity Bias.** Several prior studies [1, 16, 28, 57] have highlighted that items with higher levels of popularity are more likely to be posted and then are more frequently observed and clicked. Consequently, optimizing a ranker's performance directly on click data may result in it primarily estimating the popularity order rather than personalized relevance. As previous debiasing algorithms rely on past user feedback, particularly clicks, to estimate popularity for an item $d$, we employ the notations $(\mathcal{C}, \mathcal{O}, \mathcal{R})$ to represent the set of prior feedback (clicks, observations, relevance) associated with $d$. To illustrate, let's take $\mathcal{C}$ vs. $C$ as an example ($\mathcal{O}$ and $\mathcal{R}$ share analogous interpretations): While $C$ measures how user $u$ is likely to engage with item $d$, $\mathcal{C}$ reflects how item $d$ has been previously consumed by other users. Thus, unlike $C$, which captures the specific click value $c = 1$ assigned by user $u$ to item $d$, the

---

[*]For convenience, we only consider binary relevance here. One can easily extend our framework to a multi-level relevance case, with Eq. (28) as a possible solution to convert it to the binary setting [4, 22].

[†]We use uppercase letters (i.e., C,R,O,X) to denote random variables (see Table 1 for explanations), and lowercase letters (i.e., $c, r, o, \boldsymbol{x}$) to denote the corresponding value for each data point $d$. We further expand the random variables to represent a set of data points and use calligraphic letters (i.e., $\mathcal{C}, \mathcal{R}, \mathcal{O}$) as notation. In other words, to study the case of user $u$ and item $d$, $C = c, R = r, O = o$ where $c, r, o \in \{0, 1\}$ show whether $d$ is clicked, relevant, observed by $u$, while $\mathcal{C} = \{c = 1\}_d$, $\mathcal{R} = \{r = 1\}_d$, $\mathcal{O} = \{o = 1\}_d$ show previous users' click/relevance/observation feedback on $d$.

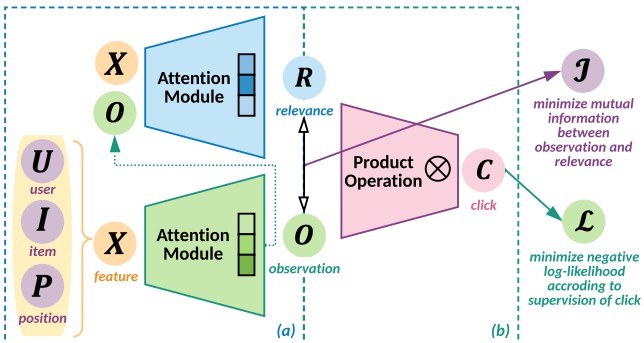

Figure 2: The overall architecture of InfoRank, where (a) we first leverage an attention mechanism to mine correlations between user-item features (Section 3.2); and (b) we then introduce a regularization formulation (i.e., $\mathcal{I}$) aimed at establishing conditional mutual information to ensure that relevance becomes conditionally independent of the observation factor (Section 3.3). To capture relevance within biased feedback, we incorporate this regularization term with supervision (i.e., $\mathcal{L}$) over user behaviors (Section 3.4). We note that InfoRank remains working even in scenarios where there is no observation information available within user browsing logs. In such cases, we substitute real observations with estimated ones.

instance of $C$ comprises the set of prior click feedback associated to $d$, denoted as $\{c = 1\}_d$. When a ranker is tasked with estimating relevance between user $u$ and item $d$, it can be considered free from popularity bias if the relevance estimation remains independent of the item's past click history. Alternatively, following the idea of collaborative filtering [40], relevance estimation could consider the item's historical relevance. This condition can be formulated as:

$$P(R = 1|C = \{c = 1\}_d, X = x) = P(R = 1|\mathcal{R} = \{r = 1\}_d, X = x). \tag{3}$$

It is worth noting that $\{c = 1\}_d$ and $\{r = 1\}_d$ do not encompass the "current" click and relevance to be estimated.

## 2.2 Causality in Ranking

Based on the above descriptions, position and popularity biases arise due to two key factors: (i) User observation feedback is influenced by the position of items. (ii) The system generates ranked lists based on observations or further clicks, which can introduce bias. Consequently, the observation factor serves as a source of bias that propagates into subsequent factors. In light of this, we consider the observation factor as the "sensitive attribute". In this regard, an ideal ranker should adhere to the following principle: for any user $u$ and item $d$, given their associated feature vector $x$, we have:

$$P(R = r|O = 1, X = x) = P(R = r|O = 0, X = x) \tag{4}$$

holds for any relevance score $r \in \{0, 1\}$, and any observation value $o \in \{0, 1\}$ attainable by $O$. We notice that Eq. (4) can be interpreted as representing the conditional independence between the latent factors $O$ and $R$. Here, external factors such as the user (or query) and item features including the position of the item are consolidated into $X$. Based on Eq. (4), one can derive the corresponding

evaluation metric as follows:

$$\Delta\text{CI} := |P(R = 1|O = 1, X = x) - P(R = 1|O = 0, X = x)|. \tag{5}$$

It is easy to show that iff Eq. (4) holds, $\Delta\text{CI} = 0$.

**Causality in Position Bias.** The fundamental approach for mitigating position bias is to estimate a relevance score $R$ that is entirely dependent on the user-item features $X$, free from any positional influence. However, we contend that simply applying Eq. (2) may not suffice to achieve this goal. This is because the estimation of an item's relevance can still be affected by whether it has been observed or not. Therefore, we advocate for an additional step to ensure the conditional independence between $R$ and $O$.

To achieve this, we combine Eq. (2) and Eq. (4) to derive:

$$P(R = 1|X = x) = P(R = 1|O = o, X = x), \tag{6}$$

where $o \in \{0, 1\}$.

**Casuality in Popularity Bias.** As per Eq. (2), we can derive $P(C = \{c = 1\}_d|X = x) = P(\mathcal{R} = \{r = 1\}_d|X = x) \cdot P(O = \{o = 1\}_d|X = x)$. It implies that given the features of an item $d$, its previous clicks (i.e., $\{c = 1\}_d$) only occur when $d$ is both relevant (i.e., $\{r = 1\}_d$) and observed (i.e., $\{o = 1\}_d$) by users. Following this, we can proceed to derive:

$$P(R = 1|C, X) = \frac{P(R = 1|O, X)}{P(R = 1|X)}P(R = 1|\mathcal{R}, X). \tag{7}$$

Here, for convenience, we use $C, O, \mathcal{R}, X$ to denote $C = \{c = 1\}_d$, $O = \{o = 1\}_d$, $\mathcal{R} = \{r = 1\}_d$, $X = x$. The detailed derivation procedure can be found in Appendix A.1.

Observing $O = \{o = 1\}_d$ and $O = 1$ are closely correlated, given that they both signify user observations, we argue that reinforcing $O = 1$ and $R = 1$'s independence conditioned on $X = x$ can lead to an approximation where $P(R = 1|O = \{o = 1\}_d, X = x)/P(R = 1|X = x)$ approaches 1. The remaining part $P(R = 1|\mathcal{R} = \{r = 1\}_d, X = x)$ reflects the ranker's inductive capacity. This capacity corresponds to the process of learning from the historical records $\mathcal{R} = \{r = 1\}_d$ to infer $R = 1$, specifically utilizing the past relevance feedback for item $d$ to infer current behavior regarding $d$.

## 2.3 Connections to Related Work

Implicit feedback such as clicks is abundant and easy to collect [24, 34, 53]. However, implicit feedback is known to be plagued by various biases [21, 25]. Previous research has predominantly focused on developing unbiased ranking methods to address position bias, which can be broadly categorized into two streams. One is built on certain assumptions about user browsing behaviors [3, 9, 15, 17, 44, 45]. These methods aim to maximize the likelihood of observed user behavior in historical data collected from browsing logs. For example, Jin et al. [24] applied survival analysis techniques to model user browsing behavior. The other follows counterfactual learning approaches [4, 27, 50] which treats click bias as a counterfactual factor [37] and mitigates user feedback biases through inverse propensity weighting [22]. For instance, Wang et al. [47] developed a new weighting scheme taking a holistic treatment of both clicks and non-clicks. Popularity bias has also been a subject of recent research. Kamishima et al. [28] introduced an information-neutral recommender by ensuring independence between the ranker outcome and viewport features. Krishnan et al.

[29] incorporated an adversarial network to play a min-max game, enabling the learning of implicit associations between popular and unpopular items.

However, each of these methods focuses on a specific bias based on a particular user browsing pattern. In contrast, our paper aims to unify these biases through a single observation factor, providing a simple and indirect means to mitigate the impact of different biases simultaneously. Furthermore, as mentioned in Section 2.2, we contend that merely "dividing" the click factor $C$ into the observation factor $O$ and the relevance factor $R$ (as done in prior approaches) is sufficient to render relevance estimations bias-free. In our approach, we go a step further and advocate for ensuring that the relevance and observation factors are conditionally independent of features.

More discussions about related casual-aware methods can be found in Appendix E.

## 3 THE INFORANK FRAMEWORK

### 3.1 Overview

As highlighted in Section 2.2, we establish the connections between conditional independence and the issues of position and popularity biases. The core idea behind InfoRank is to acquire unbiased ranking strategies by jointly optimizing the ranking performance and the achievement of conditional independence. As illustrated in Figure 2, we first use an attention mechanism to uncover latent correlations within user-item features and generate probability distributions $P(R = 1|O, X)$ and $P(O = 1|X)$ (shown in (a)). Subsequently, we translate the concept of conditional mutual dependence into a regularization term (shown as $\mathcal{I}$) and then calculate the probability distribution $P(C = 1|X)$ through a product operation (shown in (b)) according to Eqs. (2) and (6), where clicks can be supervised by biased feedback (shown as $\mathcal{L}$).

We notice that in previous literature [14, 52], relevance is estimated based on user-item features, while observation is estimated from bias-related features (e.g., position). However, we argue that the observation factor is also related to user-item features. For example, cautious users tend to thoroughly investigate various alternatives before arriving at a decision, which increases the likelihood of low-ranked items being observed. This tendency becomes even more pronounced when dealing with expensive items, as nearly everyone engages in thorough research. As the primary objective of our paper is not to manually disentangle $\boldsymbol{x}$ into two distinct components for observation and relevance estimations, we combine user-item features and positions into a unified $\boldsymbol{x}$. We then use the proposed conditional independence regularization term to derive observation and relevance estimations.

### 3.2 Unbiased Estimation

*3.2.1 Estimating $\boldsymbol{R}$ and $\boldsymbol{O}$.* A conventional method for amalgamating user-item features is to employ Multi-Layer Perceptron (MLP) layers, a technique commonly used in prior user modeling algorithms [11, 13, 39]. However, as highlighted in [35, 56], these models may lack the expressiveness required to effectively capture the intricate correlations present in user-item features. Taking Figure 1 as an instance, a teenager might prioritize price as a factor when choosing shoes due to her limited budget, indicating a correlation between age and price. This insight encourages us to adopt an approach that takes into account information from both the user and item perspectives to weigh each feature, rather than focusing solely on one side. Therefore, we incorporate a multi-head attention mechanism [42, 43]. Formally, let $\boldsymbol{x} = \{\boldsymbol{x}_0, \boldsymbol{x}_1, \ldots, \boldsymbol{x}_{N-1}\}$ denote $N$ categorical user-item features, where $\boldsymbol{x}_n$ is a one-hot *vector* representing the $n$-th feature in $\boldsymbol{x}$. In the case of non-categorical features, we directly pass them through a neural network to obtain their corresponding $\boldsymbol{x}_n$ values. Then, for the $h$-th head we have:

$$\beta_{ij}^{(h)} = (\boldsymbol{x}_i \boldsymbol{W}_T^{(h)}) \cdot (\boldsymbol{x}_j \boldsymbol{W}_S^{(h)})^{\top}, \qquad (8)$$

where $\boldsymbol{x}_i$ and $\boldsymbol{x}_j$ are the $i$-th and $j$-th feature, $\boldsymbol{W}.^{(h)}$s are trainable weights, and $\beta_{ij}^{(h)}$ determines the correlation between $\boldsymbol{x}_i$ and $\boldsymbol{x}_j$. To get a general attention value for each user-item feature, we subsequently normalize this value within the feature scope as:

$$\alpha_{ij}^{(h)} = \text{softmax}(\beta_{ij}^{(h)}) = \frac{\exp(\beta_{ij}^{(h)}/\iota)}{\sum_{j=0}^{N-1} \exp(\beta_{ij}^{(h)}/\iota)}, \qquad (9)$$

where $\iota$ denotes the temperature. We jointly attend on the feature scope from different representation subspaces to learn stably as

$$\boldsymbol{\omega}_i = \sigma\left(\boldsymbol{W}_q \cdot \left(\frac{1}{H}\sum_{h=0}^{H-1}\sum_{j=0}^{N-1}\alpha_{ij}^{(h)}(\boldsymbol{x}_j \boldsymbol{W}_C^{(h)})\right) + \boldsymbol{b}_q\right), \qquad (10)$$

where $H$ is the number of attention heads, and $\boldsymbol{W}., \boldsymbol{b}.$ are trainable parameters. We further integrate this rich information with the attention vector $\boldsymbol{w}$ to obtain feature embedding $\boldsymbol{p}$:

$$\boldsymbol{p} = \frac{1}{N}\sum_{i=0}^{N-1} \boldsymbol{w}^{\top} \cdot \tanh(\boldsymbol{W}_p \cdot \boldsymbol{\omega}_i + \boldsymbol{b}_p). \qquad (11)$$

$\boldsymbol{p}$ is further fed into two separated MLP modules activated by a sigmoid function without parameter sharing to obtain the estimation of $P(R = 1|O = o, X = \boldsymbol{x})$ and $P(O = 1|X = \boldsymbol{x})$ respectively.

**Remark.** We use Eqs. (8), (9), (10) and (11) to estimate both $P(R = 1|O = o, X = \boldsymbol{x})$ and $P(O = 1|X = \boldsymbol{x})$. The only distinction lies in the input data: when predicting the former, the input comprises the concatenation of the observation $o$ and user-item features $\boldsymbol{x}$, whereas when predicting the latter, the input includes solely the user-features $\boldsymbol{x}$. Importantly, our approach does not require the availability of raw observation data in practice, as we can generate an estimated observation variable, denoted as $\widehat{o}$, to substitute for $o$, where $\widehat{o} = 1$ when $P(O = 1|X = \boldsymbol{x}) > 0.5$, and $\widehat{o} = 0$ otherwise, as shown in Figure 2. In our experiments, all the observation information is not given.

*3.2.2 Estimating $C$.* Given the distribution $P(O = 1|X = \boldsymbol{x})$ and the conditional distribution $P(R = 1|O = o, X = \boldsymbol{x})$, we can compute $P(C = 1|X = \boldsymbol{x})$ using:

$$P(C = 1|X = \boldsymbol{x}) = P(R = 1|O = o, X = \boldsymbol{x}) \cdot P(O = 1|X = \boldsymbol{x}), \quad (12)$$

which can be derived by combining Eqs. (2) and (6).

### 3.3 Conditional Mutual Information

As discussed in Section 2.2, it is essential to ensure that relevance estimations are conditionally independent of observation estimations. This condition can be enforced through the regularization term of conditional mutual information, as per the following proposition:

PROPOSITION 3.1. *Given that relevance, click, and observation variables are binary (i.e., $R, C, O \in \{1, 0\}$), for any user-item pair with feature $X$, the following statements are equivalent:*

- *The relevance $R$ and observation $O$ are conditionally independent given $X$. In other words, $P(R, O|X) = P(R|X) \cdot P(O|X)$. That is, $P(R|O = 1, X) - P(R|O = 0, X) = 0$.*
- *The conditional mutual information between relevance $R$ and observation $O$ (later defined in Eq. (13)) is zero, i.e., $\mathcal{I}(R; O|X) = 0$.*
- *The conditional independence score $\Delta CI$ is zero.*

Please refer to the proof in Appendix A.2. PROPOSITION 3.1 allows InfoRank to minimize the conditional mutual information term $\mathcal{I}$ to indirectly enforce the conditional independence between the relevance and observation estimations.

$\mathcal{I}$ is defined as

$$\mathcal{I} := \mathcal{I}(R; O|X) = \mathbb{E}_{\boldsymbol{x} \sim \mathcal{D}} \left[ \mathcal{I}(R; O|X = \boldsymbol{x}) \right], \quad (13)$$

where

$$
\begin{aligned}
&\mathcal{I}(R; O|X = \boldsymbol{x}) \\
&= \sum_{R, O} P(R, O|X = \boldsymbol{x}) \cdot \ln \frac{P(R, O|X = \boldsymbol{x})}{P(R|X = \boldsymbol{x}) \cdot P(O|X = \boldsymbol{x})} \\
&= \sum_{R, O} P(R|O, X = \boldsymbol{x}) \cdot P(O|X = \boldsymbol{x}) \cdot \ln \frac{P(R|O, X = \boldsymbol{x})}{P(R|X = \boldsymbol{x})}.
\end{aligned}
\quad (14)
$$

Since both $P(R|O, X = \boldsymbol{x})$ and $P(O|X = \boldsymbol{x})$ can be obtained from estimations described in Section 3.2, we can then derive $P(R|X = \boldsymbol{x})$ using:

$$P(R|X = \boldsymbol{x}) = \sum_{O} P(R|O, X = \boldsymbol{x}) \cdot P(O|X = \boldsymbol{x}). \quad (15)$$

Hence, for any given user-item features $\boldsymbol{x}$, we can calculate the conditional mutual information according to Eqs. (13) (14) and (15).

## 3.4 Optimization Functions

Figure 2 illustrates our joint optimization of both $\mathcal{L}$ and $\mathcal{I}$. Regarding $\mathcal{L}$, given that click signals are binary, we employ Binary Cross Entropy (BCE) loss for click supervision. The BCE loss can be formulated as:

$$\mathcal{L} = - \sum_{(c, \boldsymbol{x}) \in \mathcal{D}} \left( c \cdot \log P(\hat{c}|\boldsymbol{x}) + (1 - c) \cdot \log(1 - P(\hat{c}|\boldsymbol{x})) \right), \quad (16)$$

where $\hat{c}$ denotes the prediction of each instance and $P(\hat{c}|\boldsymbol{x})$ is computed using Eq. (12), and $c$ is the corresponding label of the instance. In scenarios where observation information is accessible, a similar loss can be readily constructed for observation supervision. As for $\mathcal{I}$, we have derived its regularization formulation in Eq. (13).

Combining both objective functions, our aim is to minimize the log-likelihood of estimation while incorporating conditional mutual information regularization across all the data samples. The objective function is expressed as:

$$\arg\min_{\theta} \mathcal{L} + \eta \cdot \mathcal{I}, \quad (17)$$

where the hyper-parameter $\eta$ balances the prediction loss and the regularization and $\theta$ denotes all trainable parameters in InfoRank.

---

**Algorithm 1** InfoRank

---

**INPUT:** implicit feedback dataset $\mathcal{D} = \{(\boldsymbol{x}, c, o)\}$;
**OUTPUT:** unbiased ranker $f_\theta$ with parameter $\theta$

1: Initialize all parameters.
2: **repeat**
3:    Randomly sample a batch $\mathcal{B}$ from $\mathcal{D}$
4:    **for** each data point $(\boldsymbol{x}, c, o)$ in $\mathcal{B}$ **do**
5:       Calculate $P(R = 1|O = o, X = \boldsymbol{x})$ (and $P(O = 1|X = \boldsymbol{x})$) using Eqs. (8), (9), and (11).
6:       Compute $P(C = 1|X = \boldsymbol{x})$ using Eq. (12).
7:    **end for**
8:    Compute $\mathcal{L}$ and $\mathcal{I}$ according to Eqs. (16) and (13).
9:    Update $\theta$ by minimizing Eq. (17).
10: **until** convergence

---

## 3.5 Model Analysis

**Unbiased Estimation.** As discussed in Section 2.1, unbiased learning-to-rank aims to estimate unbiased relevance $r$ instead of biased click $c$, whose ideal risk function (denoted as $\widetilde{\mathcal{F}}(f)$) can be directly derived from $\mathcal{F}(f)$ (defined in Eq. (1)) by replacing $c$ with $r$.

We demonstrate that a ranker is considered unbiased if it fulfills the condition in Eq. (4), as stated in the following proposition.

PROPOSITION 3.2. *Assuming that a ranker satisfies Eq. (4), namely $P(R|O = 1, X = \boldsymbol{x}) - P(R|O = 0, X = \boldsymbol{x}) = 0$ holds and $P(O|X = \boldsymbol{x})$ is bounded away from zero for any data point in $\mathcal{D}$, then the ranker is unbiased, namely optimizing*

$$\widehat{\mathcal{F}}(f) = \sum_{u} \sum_{d \in \mathcal{D}_u} \frac{\Delta(f(\boldsymbol{x}), c)}{P(O|X = \boldsymbol{x})} \quad (18)$$

*is equivalent to optimizing $\widetilde{\mathcal{F}}(f)$.*

Please refer to the proof in Appendix A.3. It is worth noticing that Eq. (22) bears a resemblance to the risk function of inverse propensity weighting (IPW), as described in Eq. (2) in [4]. However, our approach is more comprehensive because it accounts for the influence of the observation factor to simultaneously address position and popularity biases. In a mathematical form, if Eq. (4) always holds, then our method based on Eq. (12) will simplify to an IPW-based method based on Eq. (2).

**Learning Algorithm.** We provide the learning algorithm of InfoRank in Algorithm 1. During training, we compute the overall loss (i.e., Eq. (17)) and supervise our model with click (and observation) signals; while during inference, relevance estimations $P(R|X = \boldsymbol{x})$ (which corresponds to $P(R|O = o, X = \boldsymbol{x})$ in the paper) are computed and the ranking list is generated in descending order of relevance. This process is consistent with the conventional unbiased learning-to-rank pipeline [4, 5, 22]. Additionally, although the input data of Algorithm 1 is a set of tuples $\{(\boldsymbol{x}, c, o)\}$, InfoRank can be adapted to data without any observation signal (i.e., $\{(\boldsymbol{x}, c)\}$) by replacing the label $o$ from data with the estimation $P(O|X = \boldsymbol{x})$.

**Complexity Analysis.** The main component of InfoRank is the attention module, and its complexity analysis can be summarized as follows. Let $N$ denote the total number of user-item features,

**Table 2: Comparison of different unbiased learning-to-rank methods on Yahoo Search Engine, LETOR Webpage Ranking, and Adressa Recommender System datasets. UBM is used as a click generation model. * indicates $p < 0.001$ in significance tests compared to the best baseline.**

| Ranker | Debiasing Method | Yahoo (UBM) | | | | LETOR (UBM) | | | | Adressa (UBM) | | | |
|---|---|---|---|---|---|---|---|---|---|---|---|---|---|
| | | MAP | N@3 | N@5 | N@10 | MAP | N@3 | N@5 | N@10 | MAP | N@3 | N@5 | N@10 |
| **InfoRank (Ranking)** | Labeled Data | .856 | .755 | .760 | .795 | .695 | .381 | .468 | .563 | .821 | .714 | .727 | .754 |
| | **InfoRank (Debiasing)** | **.845**\* | **.736**\* | **.739**\* | **.779**\* | **.650**\* | **.380**\* | **.460**\* | **.541**\* | **.801**\* | **.691**\* | **.715**\* | **.739**\* |
| | Regression-EM | .837 | .683 | .692 | .731 | .634 | .374 | .442 | .535 | .794 | .673 | .706 | .731 |
| | Randomization | .835 | .680 | .689 | .728 | .630 | .368 | .437 | .515 | .792 | .668 | .695 | .728 |
| | Click Data | .823 | .670 | .678 | .720 | .622 | .356 | .428 | .489 | .782 | .648 | .677 | .707 |
| LambdaMART | Labeled Data | .854 | .745 | .757 | .790 | .685 | .380 | .461 | .558 | .814 | .709 | .722 | .747 |
| | Ratio-Debiasing | .832 | .712 | .722 | .755 | .631 | .365 | .421 | .506 | .791 | .669 | .702 | .730 |
| | Regression-EM | .827 | .680 | .693 | .741 | .628 | .356 | .411 | .490 | .785 | .650 | .681 | .711 |
| | Randomization | .824 | .675 | .687 | .725 | .624 | .346 | .407 | .482 | .784 | .648 | .678 | .705 |
| | Click Data | .814 | .666 | .673 | .712 | .614 | .339 | .396 | .473 | .779 | .635 | .664 | .694 |
| DNN | Labeled Data | .831 | .685 | .705 | .737 | .678 | .364 | .454 | .551 | .802 | .700 | .722 | .745 |
| | **InfoRank (Debiasing)** | .828 | .683 | .696 | .734 | .637 | .360 | .416 | .499 | .786 | .667 | .692 | .725 |
| | Dual Learning | .825 | .680 | .693 | .730 | .625 | .352 | .410 | .487 | .784 | .663 | .688 | .720 |
| | Regression-EM | .823 | .676 | .689 | .726 | .618 | .347 | .400 | .479 | .779 | .656 | .675 | .713 |
| | Randomization | .822 | .677 | .686 | .724 | .617 | .346 | .397 | .477 | .777 | .644 | .664 | .701 |
| | Click Data | .817 | .665 | .671 | .710 | .612 | .335 | .387 | .469 | .775 | .633 | .659 | .688 |

and $H$ denote the number of attention heads. Then the time complexity of the attention model is $O(NF_1F_2H + EF_1H)$, where $E$ is the number of input feature pairs, and $F_1$ and $F_2$ are the numbers of rows and columns of the attention matrix respectively. The overall complexity of InfoRank is linear with respect to the number of features and the number of feature pairs. The prediction network's time complexity is $O(C_{att})$, where $C_{att}$ is the cost of one multi-head attention operation.

## 4 EXPERIMENTS

### 4.1 Datasets and Experimental Setting

**Data Description.** We evaluate our methods against the strong baselines over three widely adopted learning-to-rank datasets: **Yahoo search engine dataset**[‡], **LETOR webpage ranking dataset**[§], and **Adressa recommender system dataset**[¶].

**Click Data Generation.** In order to simulate the different user browsing patterns and generate click data, the following process in [4, 22, 24, 41] is followed. First, we train a Rank SVM model with 1% of training data that includes relevance labels. Second, we create an initial ranked list for each query using the trained Rank SVM model. Third, we simulate user browsing processes and sampling clicks from the initial list. To ensure that InfoRank captures diverse user browsing patterns, three simulation models are used, and each of them corresponding to a specific browsing pattern: **PBM** [36], **UBM** [15], and **CCM** [19].

We provide detailed descriptions of each dataset and each simulation model in Appendix B.1 and B.2, and hyper-parameter setting in Appendix B.3. All code will be released at publication time.

[‡]http://webscope.sandbox.yahoo.com
[§]https://www.microsoft.com/en-us/research/project/letor-learning-rank-information-retrieval/
[¶]http://reclab.idi.ntnu.no/dataset/

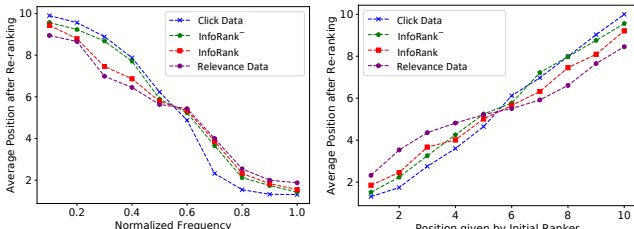

**Figure 3: Average positions after re-ranking of items at each normalized frequency (in the left subfigure); or at each original position (in the right subfigure) by different debiasing methods together with InfoRank and InfoRank⁻ on Yahoo.**

### 4.2 Baselines and Metrics

**Baseline Description.** Baselines are created by combining learning-to-rank algorithms with state-of-the-art debiasing methods. We perform comparisons against strong baselines with the debiasing methods, introduced as follows. **Randomization** [27] is a randomization technique to infer the observation probabilities. While practical, we randomly shuffled the rank lists and then estimated the position biases as in [4, 22]. **Regression-EM** [50] is a regression-based expectation maximization (EM) method, where position bias is directly estimated from regular production clicks. **Dual Learning** [4] is a dual learning algorithm, which can jointly learn a debiasing model and a ranker on click data. **Ratio-Debiasing** [22] is a pairwise unbiased learning-to-rank model based on inverse propensity weight (IPW) [49]. We use **Click Data** to represent methods directly built on raw click data without debiasing to train the ranker, whose performance is regarded as a lower bound. **Labeled Data** represents methods where a ranker is trained directly on human-annotated relevance labels without any bias, whose performance is an upper bound.

There are certain learning-to-rank algorithms that can be combined with the above debiasing methods: **DNN** is a deep neural

**Table 3: Comparison of different unbiased learning-to-rank methods under PBM, CCM. Yahoo Search Engine data and LETOR Webpage Ranking are used as click datasets. Results of Adressa Recommendation System follow similar trends, and are omitted due for brevity. * indicates $p < 0.001$ in significance tests compared to the best baseline.**

| Ranker | Debiasing Method | Yahoo (PBM) | | | Yahoo (CCM) | | | LETOR (PBM) | | | LETOR (CCM) | | |
|---|---|---|---|---|---|---|---|---|---|---|---|---|---|
| | | MAP | N@5 | N@10 | MAP | N@5 | N@10 | MAP | N@5 | N@10 | MAP | N@5 | N@10 |
| **InfoRank (Ranking)** | Labeled Data | .856 | .760 | .795 | .856 | .760 | .795 | .695 | .468 | .563 | .695 | .468 | .563 |
| | **InfoRank (Debiasing)** | **.849***| **.732***| **.772***| **.846***| **.712***| **.758***| **.681***| **.457***| **.559***| **.658***| **.455***| **.539*** |
| | Regression-EM | .841 | .715 | .740 | .822 | .685 | .734 | .675 | .453 | .552 | .652 | .450 | .534 |
| | Randomization | .840 | .704 | .736 | .817 | .679 | .728 | .671 | .450 | .551 | .649 | .449 | .531 |
| | Click Data | .831 | .682 | .725 | .808 | .658 | .710 | .647 | .445 | .510 | .640 | .439 | .498 |
| LambdaMART | Labeled Data | .854 | .757 | .790 | .854 | .757 | .790 | .685 | .461 | .558 | .685 | .461 | .558 |
| | Ratio-Debiasing | .836 | .728 | .764 | .828 | .691 | .738 | .648 | .446 | .513 | .644 | .440 | .502 |
| | Regression-EM | .830 | .700 | .743 | .816 | .675 | .727 | .636 | .436 | .509 | .634 | .431 | .497 |
| | Randomization | .827 | .690 | .728 | .814 | .673 | .722 | .633 | .433 | .498 | .628 | .427 | .493 |
| | Click Data | .820 | .672 | .716 | .804 | .653 | .706 | .630 | .424 | .494 | .625 | .418 | .488 |
| DNN | Labeled Data | .831 | .705 | .737 | .831 | .705 | .737 | .678 | .454 | .551 | .678 | .454 | .551 |
| | **InfoRank (Debiasing)** | .829 | .703 | .736 | .828 | .692 | .735 | .651 | .446 | .547 | .650 | .444 | .531 |
| | Dual Learning | .828 | .697 | .734 | .823 | .681 | .731 | .645 | .437 | .528 | .638 | .430 | .525 |
| | Regression-EM | .829 | .699 | .736 | .819 | .678 | .728 | .635 | .426 | .500 | .628 | .417 | .490 |
| | Randomization | .825 | .693 | .732 | .816 | .674 | .726 | .630 | .419 | .495 | .625 | .415 | .487 |
| | Click Data | .819 | .667 | .711 | .801 | .650 | .705 | .629 | .419 | .492 | .621 | .409 | .480 |

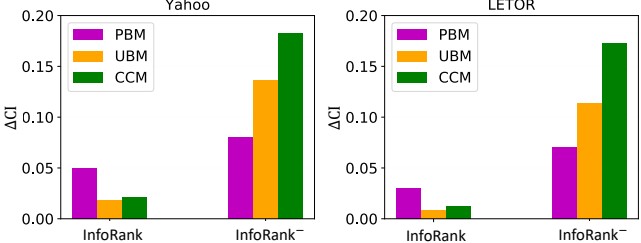

**Figure 4: Comparison of InfoRank and InfoRank⁻ under different click generation models and datasets in terms of the $\Delta$CI metric.**

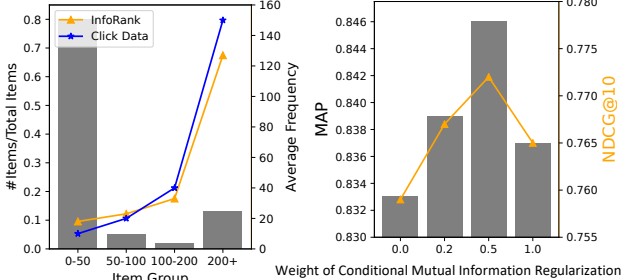

**Figure 5: Left: Average frequency of different item groups recommended by InfoRank (Ranking) incorporated with InfoRank (Debiasing) and Click Data on Adressa. Right: Performance change of InfoRank with different regularization weight $\eta$ on Yahoo.**

network ranker as described in [4]. **LambdaMART** is a widely used learning-to-rank algorithm as described in [6].

Let **InfoRank** denote ours, where InfoRank (Debiasing) denotes the proposed debiasing method based on the conditional mutual information minimization and InfoRank (Ranking) denotes the proposed attention-based learning-to-rank method. We introduce InfoRank⁻ to denote the variant of InfoRank without the conditional mutual information regularization.

Following the experimental settings in [22], we do not incorporate Ratio-Debiasing with DNN and Dual Learning with LambdaMART, since DNN is originally designed for Dual Learning and Ratio-Debiasing is typically designed based on LambdaMART. We also evaluate the performance of incorporating InfoRank (Debiasing) into DNN by using DNN as our estimations instead of the proposed attention network.

**Evaluation Metrics.** Evaluation metrics chosen for assessing performance are NDCG at positions 3, 5, 10 (denoted as N@3, N@5, N@10) and MAP at position 10 (denoted as MAP) as evaluation metrics. Additionally, the paper introduces another evaluation metric, denoted as $\Delta$CI in Eq. (5) to measure the casual dependence defined in Section 2.2.

## 4.3 Performance Comparison

We summarize the results regarding the ranking performance in Table 2. The results of Ratio-Debiasing, Regression-EM and Dual Learning are comparable with those reported in [22]. We show the major findings as follows:

- Our model that combines InfoRank (Ranking) and InfoRank (Debiasing) together achieves better performance than any combinations of the existing state-of-the-art ranker and debiasing methods (e.g., the combination of LambdaMART with Ratio-Debiasing, the combination of DNN with Dual Learning) in terms of all measures, which verifies the superiority of our framework.
- We find that although sophisticated ranking algorithms like InfoRank (Ranking), LambdaMART and DNN can achieve good performance on click data, they are sensitive to position bias when comparing against the performance of debiasing methods, which indicates the importance of unbiased learning-to-rank.
- When using the same ranker, InfoRank (Debiasing) consistently outperforms other debiasing methods; and when under the same

debiasing method (e.g., Regression-EM, Randomization), InfoRank (Ranking) could achieve the best performance.

## 4.4 Debiasing Analysis

**Visualization Analysis of Popularity Bias.** For popularity bias, since there is no unique ID for each item, we first adopt the K-means algorithm where item similarity is measured by Pearson's coefficient. We use the query number as the number of classes, and regard items in the same class as the same item. We then study the correlations between the item frequency and the position of re-ranking lists. As depicted in Figure 3(Left), the curve of click data (in blue with mark "×") is far from that of relevance labels (in purple with mark '•'), indicating that directly using click data without debiasing can be problematic. The curve of InfoRank (in red with "■") is the closest to the relevance label curve, indicating that the performance enhancement of InfoRank is indeed, at least partly, due to effective debiasing.

**Visualization Analysis of Position Bias.** For position bias, we compare the ranking list given by the debiased ranker against the initial one. Specifically, we first identify the items at each position given by the initial ranker. Then we calculate the average positions of items at each original position after re-ranking by various debiasing methods, combined with InfoRank. We also calculate the their average positions after re-ranking by their relevance labels, which is regarded as the ground-truth. Ideally the average positions produced by the debiasing methods should be close to the average position by relevance labels. Similar to the visualization analysis of popularity bias, Figure 3(Right) shows that the curve of InfoRank (in red with "■") is the closest to the relevance label curve, revealing that InfoRank could truly decline the effect of position bias.

**Ablation Study on ΔCI Metric.** We further conduct evaluations under different simulators and datasets in terms of the ΔCI metric. From results shown in Figure 4, we can see that InfoRank with the conditional mutual information regularization works better than InfoRank without it in terms of ΔCI metric. We also notice that the regularization term shows weaker impact on PBM than UBM and CCM. One explanation is that PBM directly assumes the observation is solely determined by the position instead of user-item feature, namely the observation is naturally independent of the relevance. By contrast, given the formulation of UBM and CCM, the observation is dependent on the click, and thus further is influenced by user-item feature.

**Ablation Study of Item Groups with Different Frequency.** We follow [51] to further investigate whether InfoRank alleviates the popularity bias issue. Table 2 shows InfoRank (Debiasing) significantly outperforms Click Data, and Figure 5(Left) shows the recommendation frequency of popular items is reduced. It means that the models trained by Click Data are prone to recommend more popular items to unrelated users due to popularity bias. In contrast, InfoRank (Debiasing) reduces the item's direct effect and recommends popular items mainly to suitable users.

**Hyperparameter Study of $\eta$.** To evaluate how the weight of the regularization $\eta$ influences the performance of InfoRank, we set $\eta$ = 0.0, 0.2, 0.5, 1.0 and test the performance on Yahoo. Results depicted

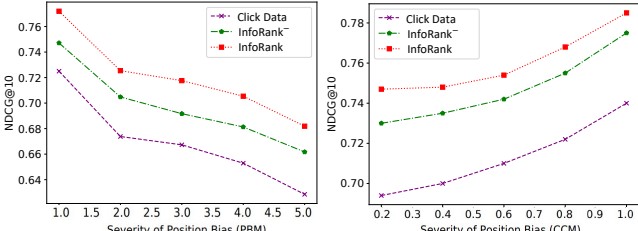

**Figure 6: Performance of InfoRank against click data with different degrees of position bias on Yahoo.**

in Figure 5(Right) imply that the proposed mutual information minimization can enhance the ranking performance.

## 4.5 Robustness Analysis

**Generalizability to Different Click Generations.** To test the generalizability of InfoRank, we use three different click generation models to generate data and conduct comparison experiments against the baselines on these data. Each click model simulates a certain user browsing pattern. InfoRank has strong performance on data generated with UBM (as shown in Table 2), and also outperforms baselines on the data generated from PBM and CCM (as shown in Table 3), which shows that the idea of InfoRank is general to various user patterns.

**Different Degrees of Position Bias.** In the above experiments, we only test the performance of InfoRank and InfoRank⁻ with click data generated from click models with a given degree of position bias, i.e., $\gamma_1 = 0.5$ in CCM and $\tau = 1$ in PBM. Here, $\gamma_1$ and $\tau$ influence the probability of a user examining the next result. Smaller $\gamma_1$ and larger $\tau$ indicate a smaller probability to continue reading, which means a more severe position bias. Therefore, we set the hyper-parameters for each click generation model to five values and examine whether InfoRank is still equally effective. The left and right subfigures of Figure 6 show the NDCG@10 results as the degree of position bias increases; the results in terms of other measures follow similar trends. When $\tau$ in PBM is 0, there is no position bias; while $\gamma_1$ in CCM is 1, there still exists position bias from $\gamma_2$ and $\gamma_3$. As we add more position bias, i.e., as $\tau$ increases and $\gamma_1$ decreases, the performance of all the debiasing methods decreases dramatically. One can see that under all these settings, InfoRank is less affected by position bias and consistently maintains the best results.

We include a robustness analysis for varying amounts of training data in Appendix C and we also assess the feasibility of deployment in Appendix D.

## 5 CONCLUSION AND FUTURE WORK

In this paper, we propose a novel unbiased learning-to-rank framework named InfoRank. Our fundamental insight is to consolidate various biases into a unified observation factor, allowing us to learn a precise ranker while effectively mitigating position and popularity biases. For future endeavors, it would be intriguing to extend the applicability of InfoRank to address additional ranking biases.

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

# APPENDIX

## A PROOFS AND DEVIATIONS

### A.1 Detailed Deviation of Eq. (7)

We begin by extending Eq. (2) to:

$$P(C = \{c = 1\}_d|X) = P(\mathcal{R} = \{r = 1\}_d|X) \cdot P(O = \{o = 1\}_d|X), \tag{19}$$

which means that given the feature of an item $d$, then its previous clicks (i.e., $\{c = 1\}_d$) only occur when $d$ is both relevant $\{r = 1\}_d$ and observed (i.e., $\{o = 1\}_d$) by users.

$$\begin{aligned} P(R|C, X) &\stackrel{(1)}{=} \frac{P(C|R, X)P(R|X)}{P(C|X)} \\ &\stackrel{(2)}{=} \frac{P(O|R, X)}{P(O|X)} \frac{P(\mathcal{R}|R, X)P(R|X)}{P(\mathcal{R}|X)} \\ &\stackrel{(3)}{=} \frac{P(O|R, X)P(R|X)}{P(O|X)P(R|X)} P(R|\mathcal{R}, X) \\ &\stackrel{(4)}{=} \frac{P(R|O, X)}{P(R|X)} P(R|\mathcal{R}, X), \end{aligned} \tag{20}$$

where for simplicity, we use $C$, $O$, and $\mathcal{R}$ to denote $C = \{c = 1\}_d$, $O = \{o = 1\}_d$ and $\mathcal{R} = \{r = 1\}_d$. Easy to see there are four steps in the above equation, where steps 1,3,4 use Bayes' theorem and step 2 uses Eq. (19). $C, O, \mathcal{R}$ are random variables whose meanings are extended but different from $C, O, R$.

The instance of $C, O, R$ is the corresponding value $c, o, r$ (from user $u$) for each data point $d$, while the instance of $C, O, \mathcal{R}$ is a set of previous $c, o, r$ associated with $d$, denoted as $\{c\}_d, \{o\}_d, \{r\}_d$. Since variables $C, O, \mathcal{R}$ are expressed in different representation spaces from $C, O, R$, it is not practical to directly enforce the independence of $O$ and $R$ conditioned on $X$. However, it is observed that $O$ and $O$ exhibit a strong correlation since they both represent user observations. As a result, reinforcing the independence of $O$ and $R$ conditioned on $X$ can approximate the ratio $P(R|O, X)/P(R|X)$ to be close to 1. The variables $\mathcal{R}$ and $R$ exist in different representation spaces but represent the same factor, i.e., relevance. Therefore, $P(R|\mathcal{R}, X)$ indicates the inductive ability of the ranker, which corresponds to the process of learning from historical relevance values $\mathcal{R}$ to predict the current relevance value $R$.

### A.2 Proof of Proposition 3.1

PROPOSITION A.1. *Given that relevance, click, and observation variables are binary (i.e., $R, C, O \in \{1, 0\}$), for any user-item pair with feature $X$, the following statements are equivalent:*

1. *The relevance $R$ and observation $O$ are conditionally independent given $X$. In other words, $P(R, O|X) = P(R|X) \cdot P(O|X)$. That is, $P(R|O = 1, X) - P(R|O = 0, X) = 0$.*
2. *The conditional mutual information between relevance $R$ and observation $O$ (later defined in Eq. (13)) is zero, i.e., $\mathcal{I}(R; O|X) = 0$.*
3. *The conditional independence score $\Delta CI$ is zero.*

PROOF. By combining Eqs. (4) and (5), it is trivial to conclude that *statements* 1 and 3 are equivalent.

Here, we further prove that *statements* 1 and 2 are equivalent. According to Eq. (14), *statement* 2 implies $P(R|O = 1, X) = P(R|O = 0, X)$. Given that $O \in \{0, 1\}$, we can derive that $P(R|O, X) = P(R|O = 0$

$o, X)$ which holds for all $o \in \{0, 1\}$. Hence, we have:

$$\begin{aligned} P(R|X) &= \sum_O P(R|O, X) \cdot P(O|X) \\ &= P(R|O = o, X) \cdot \sum_O P(O|X) \\ &= P(R|O, X), \end{aligned} \tag{21}$$

from which we can further derive: $P(R, O|X) = P(R|O, X) \cdot P(O|X) = P(R|X) \cdot P(O|X)$. From the analysis above, we can derive *statement* 1 from *statement* 2. Also, we can obtain *statement* 2 from *statement* 1 in a similar way. □

### A.3 Proof of Proposition 3.2

PROPOSITION A.2. *Assuming that a ranker satisfies Eq. (4), namely $P(R|O = 1, X = \boldsymbol{x}) - P(R|O = 0, X = \boldsymbol{x}) = 0$ holds and $P(O|X = \boldsymbol{x})$ is bounded away from zero for any data point in $\mathcal{D}$, then the ranker is unbiased, namely optimizing*

$$\widehat{\mathcal{F}}(f) = \sum_u \sum_{d \in \mathcal{D}_u} \frac{\Delta(f(\boldsymbol{x}), c)}{P(O|X = \boldsymbol{x})} \tag{22}$$

*is equivalent to optimizing $\widetilde{\mathcal{F}}(f)$.*

PROOF. We can obtain the formulation of the risk function of unbiased ranker from Eq. (1) as

$$\widetilde{\mathcal{F}}(f) = \int_u \int_{d \in \mathcal{D}_u} \Delta(f(\boldsymbol{x}), r) \, dP(\boldsymbol{x}, r). \tag{23}$$

However, $\widetilde{\mathcal{F}}(f)$ cannot be computed directly, and is typically estimated via the following empirical risk function:

$$\mathcal{F}'(f) = \frac{1}{|\mathcal{D}|} \sum_u \sum_{d \in \mathcal{D}_u} \Delta(f(\boldsymbol{x}), r), \tag{24}$$

where $\Delta$ denotes a point-wise loss function. According to Eq. (12), we can derive that

$$\begin{aligned} \mathbb{E}(\widehat{\mathcal{F}}(f)) &= \mathbb{E}\left(\sum \frac{\Delta(f(\boldsymbol{x}), c)}{P(O|X = \boldsymbol{x})}\right) \\ &= \mathbb{E}\left(\sum \frac{o \cdot \Delta(f(\boldsymbol{x}), r')}{P(O|X = \boldsymbol{x})}\right) \\ &= \sum \left(\mathbb{E}(o) \frac{\Delta(f(\boldsymbol{x}), r')}{P(O|X = \boldsymbol{x})}\right) \\ &= \sum \left(P(O|X = \boldsymbol{x}) \frac{\Delta(f(\boldsymbol{x}), r')}{P(O|X = \boldsymbol{x})}\right) \\ &= \sum \Delta(f(\boldsymbol{x}), r') = \mathcal{F}''(f), \end{aligned} \tag{25}$$

where we omit $\frac{1}{|\mathcal{D}|}$ since it is a constant and use $\sum$ to denote $\sum_u \sum_{d \in \mathcal{D}_u}$ for convenience.

Here, $r$ in Eq. (24) is estimated with $P(R|X = \boldsymbol{x})$, while $r'$ in Eq. (25) is estimated with $P(R|O = o, X = \boldsymbol{x})$. Since the ranker satisfies Eq. (4), according to Eq. (6) we have:

$$P(R|X = \boldsymbol{x}) = P(R|O = o, X = \boldsymbol{x}). \tag{26}$$

So optimizing $\mathcal{F}''(f)$ is equivalent to optimizing $\mathcal{F}'(f)$, implying InfoRank is an unbiased estimator.

Note that we here share some equations with [27]. □

## B  EXPERIMENTAL CONFIGURATION

### B.1  Dataset Description

We evaluate our methods against the strong baselines over three widely adopted learning-to-rank datasets.

- **Yahoo search engine dataset** is one of the largest benchmark datasets widely used in unbiased learning-to-rank [5, 22, 24]. It consists of 29,921 queries and 710k documents. Each query-document pair is represented by a 700-dimensional feature vector manually assigned with a label denoting relevance at 5 levels [8].
- **LETOR webpage ranking dataset** is a package of benchmark datasets for research on LEarning TO Rank, which uses the Gov2 web page collection and two query sets from the Million Query track of TREC2007 and TREC2008. We conduct experiments on MQ2007, one of two datsets for supervised learning-to-rank tasks in LETOR. MQ2007 contains 2,476 queries and 85K documents. Each query-document pair is represented by a 46-dimensional feature vector with a manually assigned 3-level label [33].
- **Adressa recommender system dataset** is a news dataset that includes news articles in connection with anonymized users, where we can regard user and item information as query and document features respectively. Similar approaches can be found in [24, 46]. It contains 2,287K historical logs of 561,733 users and 11,207 articles. Each query-document pair is represented with several contextual attributes, such as title and category [18]. Following [7], we first calculate the normalized reading time $nv_{u,d}$ with respect to both users' reading habits and the length of different articles:

$$nv_{u,d} = \frac{v_{u,d}}{\mu_d},\qquad(27)$$

where $v_{u,d}$ is the time user $u$ spent reading article $d$, and $\mu_d$ is the average reading time for article $d$. Then we convert all $nv_{u,d}$ to binary relevance by mapping the top 20% to 1 and the rest to 0.

Let $y$ represent the relevance level in each dataset, then $y \in \{0,1\}$ for Adressa Recommender system dataset, $y \in \{0,1,2\}$ for LETOR webpage ranking dataset, and $y \in \{0,1,2,3,4\}$ for Yahoo search engine dataset. Following [4, 22], we transform the relevance into a binary space by assigning a threshold on the probability of relevance $P(r = 1)$ which is calculated by

$$P(r = 1) = \epsilon + (1 - \epsilon) \cdot \frac{2^y - 1}{2^{y_{\max}} - 1},\qquad(28)$$

where $\epsilon$ denotes click noise and is set to 0.1 by default.

For all datasets, we remove sequences whose relevance is all 0. We also remove extremely long sequences, namely those that contain more than 50 events.

### B.2  Click Data Generation

In order to generate the click data from the relevance data, we employ the following three click generators, each of which is designed for a certain user browsing patterns.

- **PBM** [36] simulates user browsing behavior based on the assumption that the bias of an item only depends on its position, which can be formulated as $P(o_i) = \rho_i^\tau$, where $\rho_i$ represents position bias at position $i$ and $\tau \in [0, +\infty]$ is a parameter controlling the degree of position bias. The position bias $\rho_i$ is obtained from an eye-tracking experiment in [26] and the parameter $\tau$ is set as 1 by

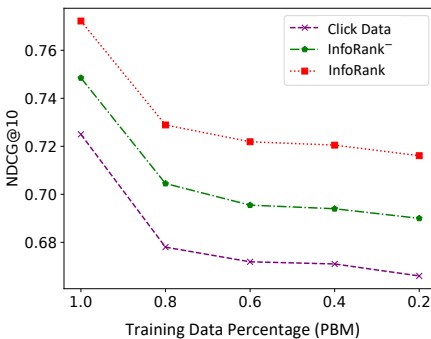

**Figure 7: Performance of InfoRank and InfoRank$^-$ against click data with different amounts of training data on Yahoo.**

default. It also assumes that a user decides to click a document $d_i$ according to the probability $P(c_i) = P(o_i) \cdot P(r_i)$.

- **UBM** [15] is an extension of the PBM model that has some elements of the cascade model. The examination probability depends not only on the rank of an item $d_i$ but also on the rank of the previously clicked document $d_{i'}$ as $P(o_i = 1|c_{i'} = 1, c_{i'+1} = 0, \ldots, c_{i-1} = 0) = \gamma_0$. Similarly, we get $\gamma_0$ from the eye-tracking experiments in [15, 26]. The click probability is determined by $P(c_i) = P(o_i) \cdot P(r_i)$.
- **CCM** [19] is a cascade model, which assumes that the user browses search results in a sequential order from top to bottom. User browsing behavior is conditioned on both current and past documents, as $P(c_i = 1|o_i = 0) = 0$, $P(c_i = 1|o_i = 1, r_i) = P(r_i)$, $P(o_{i+1} = 1|o_i = 0) = 0$, $P(o_{i+1} = 1|o_i = 1, c_i = 0) = \gamma_1$, $P(o_{i+1} = 1|o_i = 1, c_i = 1, r_i) = \gamma_2 \cdot (1 - P(r_i)) + \gamma_3 \cdot P(r_i)$. The parameters are obtained from an experiment in [19] with $\gamma_2 = 0.10$ and $\gamma_3 = 0.04$ for navigational queries (Yahoo and LETOR datasets); $\gamma_2 = 0.40$ and $\gamma_3 = 0.27$ for informational queries (Adressa dataset). $\gamma_1$ is set to 0.5 by default.

### B.3  Hyperparameter Setting

Our experiments were implemented with Tensorflow. From the baseline methods, we directly use their defacult hyper-parameters. For InfoRank, we set the learning rate as 0.001, batch size as 128, batch normalization decay as 0.9, the weight of $\alpha$ as 0.01, L2 regularization weight as 0.01, and optimize with Adam. In the main experiment, we set $\eta$ as 0.5. All the models are trained under the same hardware settings with 16-Core AMD Ryzen 9 5950X (2.194GHZ), 62.78GB RAM, NVIDIA GeForce RTX 3080 cards.

## C  ADDITIONAL EXPERIMENTAL RESULTS

**Different Amounts of Training Data.** We study the robustness of InfoRank with or without the conditional mutual information regularization, denoted as InfoRank or InfoRank$^-$, under different amounts of training data on Yahoo. We first randomly select a subset of training data (i.e., 20% - 100%) to generate click data, and then use these datasets to train InfoRank with different debiasing methods. For fair comparisons, we use the same data for evaluation across all experiments. As shown in Figure 7, when the amount of training data decreases, the improvements obtained by the debiasing methods also decrease. The reason could be that the position

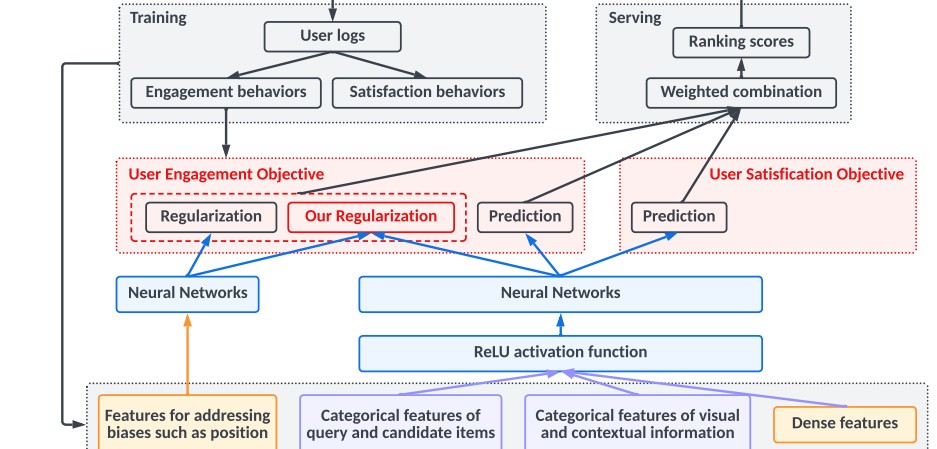

**Figure 8: Online unbiased learning-to-rank system with the proposed InfoRank model. The new system plugins the proposed conditional mutual information regularization in the industrial pipeline described in [55].**

bias estimated from insufficient training data is not accurate, which can hurt the performance of debiasing methods.

## D  DEPLOYMENT FEASIBILITY

We are actively pursuing an opportunity to deploy InfoRank in the operational schedule of daily item recommendation platform within a mainstream e-commerce company. In this context, we would like to discuss the feasibility of InfoRank's industrial deployment.

The transition from the current unbiased learning-to-rank model pipeline to InfoRank involves some key considerations. The main change brought about by InfoRank is the computation and minimization of conditional mutual information. To update the model pipeline to InfoRank, we would need to incorporate an estimator for observation in addition to the relevance estimator and perform the computation of conditional mutual information. Fortunately, many existing unbiased search engines (e.g., [55]) have already implemented a shadow tower to model the influence of ranking biases, especially position bias. In most cases, this shadow tower is responsible for generating observation estimations, as position significantly affects the observation of an item. Therefore, to integrate the proposed conditional mutual information minimization into the pipeline, we only need to adapt their regularization term to ours, as outlined in Eq. (13). This modification mainly involves switching from the existing "regularization" term (shown in the black box in Figure 8) to our proposed "conditional mutual information regularization" term (shown in the red box in Figure 8).

Efficiency is another essential concern in industrial applications. We have conducted an analysis of the time complexity of InfoRank in Section 3.5. Importantly, adding the new regularization term has minimal impact on computational cost. Consequently, the overall increase in computational load is mostly attributable to Eq. (13), resulting in only a slight additional cost in terms of complexity and almost no discernible effect on system performance.

Moreover, it is important to mention that InfoRank, following a two-tower architecture, can not be reduced to a single-tower architecture, even with accessible observation information. This architectural choice is made to introduce inductive biases to enable more efficient use of limited observational logs and improve generalization.

## E  ADDITIONAL RELATED WORK

In Section 2.3, we can summarize the difference between our method and IPW-based debiasing methods. Recently, casual-aware methods have thrived in analyzing the root causes of bias problems [12, 20, 30, 38, 48, 54] and applying backdoor adjustment [32] during training or inference to address the bias problems. For example, Zhang et al. [54] identifies popularity as a confounder that affects both item exposures and user clicks. However, the key objective of our paper is not to discover confounders between the recommended items and corresponding user feedback. Instead, we directly estimate relevance and observation factors to formulate their relationship and enhance the independence between observation and relevance estimations.

