# OpenReview forum: "InfoRank: Unbiased Learning-to-Rank via Conditional Mutual Information Minimization"
_ACM.org/TheWebConf/2024/Conference — TheWebConf24_

### Official Review · Reviewer_gf6H · 2023-11-22

**Novelty:** 3
**Technical Quality:** 4

**Review:**

The paper tackles the important problem of position and popularity biases in recommendation systems. The paper unifies those two biases into a single observation factor, and debiases it. In particular, the paper learns an end-to-end click model that can be separated as a product of two attention-based binary classifiers: observation model and relevance model. Only if the classifiers predict both observation and relevance, then a click is predicted. Finally, the paper suggests the resulting relevance model as the unbiased ranker. The paper tests the solution on synthetic data, outperforms several baselines.

Presentation: I found the paper well written and well motivated.

Novelty: The paper's main claim-to-fame is providing a unified approach to deal with both position and popularity biases. However, I do not believe this unified approach is new. Several papers designed a unified approach to deal with several biases, including popularity and selection biases (For example, AutoDebias by Chen et al 2021, debias popularity bias as part of the exposure bias). In addition, the idea of learning through parallel observation and relevance models such that click is the "and" operation of them is well known in debiasing. However, I don't remember seeing attention-based models with the same input used for this task - if this is the main novelty of the paper I suggest emphasizing this point.

Experiment: The paper demonstrates the power of its method on syntactic data using several observation models, and shows meaningful results. I find the use of several observation models important to convince that in a real-life scenario the method will also work well. However, and I know it is challenging to perform, there is no replacement for an online experiment in debiasing. Finally, I find the baselines relatively weak, as no baseline method from the last 4 years was used. Since I view the main novelty as attention-based architecture, I believe it should be compared to recent works.

In conclusion, the paper has its merits, but I believe the main contribution should be restated, and additional baselines should be added to the experiments.

**Questions:**

What is the novelty of the paper? Is it indeed the use of attention-based architecture, or is it the unified approach for debiasing several types of biases? If it is the latter, please explain the difference from previous works.

**Ethics Review Description:**

-

**Reviewer Confidence:**

3: The reviewer is confident but not certain that the evaluation is correct

**Scope:**

4: The work is relevant to the Web and to the track, and is of broad interest to the community

---

### Official Review · Reviewer_yuH2 · 2023-11-23

**Novelty:** 2
**Technical Quality:** 3

**Review:**

#### [Summary]
This paper targets the unbiased learning-to-rank problem, with a particular focus on position and popularity bias. This paper proposes a new framework called InfoRank. InfoRank employs two separate MLP modules: one to estimate relevance given the observation and user-item features, and the other to estimate the observation given the user-item features. The click is then estimated by multiplying the outputs from these MLP modules. The InfoRank framework learns using binary cross-entropy loss with ground-truth click data, along with a regularization term based on conditional mutual information. The experiments are conducted on three ranking models (i.e., DNN, LambdaMART, and InfoRANK) and on three datasets (i.e., Yahoo, LETOR, and Adressa).


#### [Strengths]
- This paper targets an important research problem for the ranking model.
- The proposed method is explained in detail and is easy to understand.
- This paper provides experimental results with statistical significance tests.


#### [Weaknesses]
- My major concern pertains to the novelty of this paper. The major distinction of InfoRank is to promote conditional independence of relevance and observation with two separate MLP modules. The idea of decomposing implicit user feedback into observation and relevance has been extensively studied in the literature, often estimated by independent models [1]. Furthermore, the conditional independence of relevance and observation, also referred to as the unconfoundedness assumption, has recently been questioned by researchers [2]. In this context, I believe that the novelty and impact of the proposed framework are limited.

    - [1] Unbiased Recommender Learning from Missing-Not-At-Random Implicit Feedback, WSDM'20

    - [2] Reconsidering Learning Objectives in Unbiased Recommendation with Unobserved Confounders, KDD'23



- I believe this paper overlooks a very closely related work [3] that also decouples the effects of relevance and observation for unbiased learning to rank problem. I recommend that the authors discuss and compare this work in the paper.

    - [3] LBD: Decouple Relevance and Observation for Individual-Level Unbiased Learning to Rank, NeurIPS'22

- In line 446, isn't it "user-item features", instead of "user-features"?

**Questions:**

Please refer to the weaknesses described in my review. Thank you.

**Ethics Review Description:**

I don't have ethical concerns

**Reviewer Confidence:**

3: The reviewer is confident but not certain that the evaluation is correct

**Scope:**

4: The work is relevant to the Web and to the track, and is of broad interest to the community

---

### Official Review · Reviewer_9oe5 · 2023-11-28

**Novelty:** 5
**Technical Quality:** 5

**Review:**

The paper discusses the challenges associated with ranking items based on user interests, particularly the biases introduced by past click-through behaviors. To address these biases, the paper proposes a new learning-to-rank paradigm called InfoRank. InfoRank aims to simultaneously handle position and popularity biases by consolidating them into a unified observation factor. The approach involves minimizing the mutual information between observation and relevance estimations conditioned on input features, ensuring bias-free relevance estimation. Implementation includes an attention mechanism to capture latent correlations within user-item features and a regularization term based on conditional mutual information to promote conditional independence. Experimental evaluations on three extensive recommendation and search datasets demonstrate that InfoRank produces more precise and unbiased ranking strategies.

The work is timely and relevant to the WebConf

**Questions:**

Baselines used are strong. However, i wonder why the baselines in references 24, 47 and 28 are not used especially your approach is unifying these biases.

**Ethics Review Description:**

dataset based work and hence no obvious issues noticed

**Reviewer Confidence:**

2: The reviewer is willing to defend the evaluation, but it is likely that the reviewer did not understand parts of the paper

**Scope:**

4: The work is relevant to the Web and to the track, and is of broad interest to the community

---

### Official Review · Reviewer_dFXv · 2023-11-29

**Novelty:** 5
**Technical Quality:** 5

**Review:**

Summary:
The paper titled "InfoRank: Unbiased Learning-to-Rank via Conditional Mutual Information Minimization" addresses the challenge of bias in learning-to-rank systems, specifically focusing on position and popularity biases. Learning-to-rank is crucial in applications like recommender systems, where user feedback (such as clicks) is used to rank items. However, this feedback is often biased towards items that are already ranked highly, creating a "rich-get-richer" effect. The paper proposes InfoRank, a novel paradigm that aims to simultaneously address both position and popularity biases by consolidating these biases into a single 'observation' factor. The approach involves minimizing mutual information between observation estimation and relevance estimation, conditioned on input features, to ensure unbiased relevance estimation. InfoRank uses an attention mechanism to capture latent correlations in user-item features and introduces a regularization term based on conditional mutual information. This framework was tested across three diverse datasets, demonstrating its effectiveness compared to state-of-the-art baselines.

Strengths:
1. InfoRank introduces a unique method for addressing biases in learning-to-rank systems. The consolidation of biases into a single observation factor and the use of conditional mutual information for debiasing are novel and potentially impactful contributions.
2. The framework was evaluated across three diverse datasets, ensuring that the findings are robust and applicable across different scenarios.
3.  The paper suggests that the conditional mutual information minimization approach can potentially enhance other ranking models, indicating a broader impact beyond the specific framework of InfoRank.

Weaknesses:
1. The method, while effective, appears complex in terms of implementation. This complexity might limit its adoption in practical settings where simpler solutions are preferred.
2. Although the paper shows effectiveness on diverse datasets, real-world applicability and performance in live environments have not been demonstrated.
3. The reliance on an attention mechanism and multiple layers of modeling could lead to overfitting, especially in scenarios with limited or noisy data.

**Questions:**

1. How does InfoRank perform in real-world, live environments, especially where user behavior and item popularity can be highly dynamic?
2. Is there a risk of overfitting due to the complexity of the model, and how does InfoRank address this?
3. Can the techniques used in InfoRank be simplified for easier implementation without significantly compromising on performance?

**Reviewer Confidence:**

3: The reviewer is confident but not certain that the evaluation is correct

**Scope:**

4: The work is relevant to the Web and to the track, and is of broad interest to the community

---

### Official Review · Reviewer_uAZq · 2023-12-01

**Novelty:** 5
**Technical Quality:** 4

**Review:**

This paper proposes a new unbiased learning-to-rank model, named InforRank, via conditional mutual information minimization. The model is interesting and clear introduced. Various experiments are conducted to show its effectiveness.

Some Strengths:
1. This paper focuses on unbiased learning to rank, a classic and important research problem in information retrieval scenarios.
2. The method section of the paper is detailed, and the appendix also includes a complete proof process.
3. The author conducted various experiments to verify the effectiveness of the proposed model across multiple datasets with different settings.

However, there are also some weaknesses:
1. In the experimental section, the paper does not compare the proposed method with state-of-the-art related work, even though the author cited some recent related works [30].
2. The introduction of recent works in the related work section is insufficient, citing only one paper from 2022 and one from 2023.

**Questions:**

1. Concerns about the choice of baselines.
2. In Tables 2 and 3, did the author compare whether the debiasing strategy of InfoRank achieves the best performance under different ranker settings?

**Ethics Review Description:**

No ethical issues

**Reviewer Confidence:**

2: The reviewer is willing to defend the evaluation, but it is likely that the reviewer did not understand parts of the paper

**Scope:**

4: The work is relevant to the Web and to the track, and is of broad interest to the community

---

### Decision · Program_Chairs · 2024-01-22

**Decision:**

Accept

**Comment:**

This paper focuses on learning-to-rank debiasing by decomposing the relevance and observation factors. The topic is highly relevant to the search track. The key contribution include an attention mechanism to capture hidden correlations among user-item features, and the regularization term to guide the disentanglement. These two factors are mainly overlooked by the existing literature. The raised concerns include the unclear motivation, lack of more baselines and limited analysis. Most of these concerns are well addressed in the author rebuttal. The rest would also be well fixed after simple revise. Overall, the pros seem to outweigh the cons, and an acceptance would be delivered given there are sufficient slots.